# Selective Arterial Embolization of Ruptured Hepatocellular Carcinoma with N-Butyl Cyanoacrylate and Lipiodol: Safety, Efficacy, and Short-Term Outcomes

**DOI:** 10.3390/jpm13111581

**Published:** 2023-11-07

**Authors:** Jory Cali, Olivier Chevallier, Kévin Guillen, Marianne Latournerie, Amin Mazit, Ludwig Serge Aho-Glélé, Romaric Loffroy

**Affiliations:** 1Department of Vascular and Interventional Radiology, François-Mitterrand University Hospital, 14 Rue Paul Gaffarel, BP 77908, 21000 Dijon, France; jory.cali@chu-dijon.fr (J.C.); olivier.chevallier@chu-dijon.fr (O.C.); amin.mazit@chu-dijon.fr (A.M.); 2ICMUB Laboratory, UMR CNRS 6302, University of Burgundy, 9 Avenue Alain Savary, 21000 Dijon, France; 3Department of Gastroenterology and Hepatology, Francois-Mitterrand University Hospital, 14 Rue Paul Gaffarel, BP 77908, 21000 Dijon, France; marianne.latournerie@chu-dijon.fr; 4Department of Epidemiology, Statistics and Clinical Research, François-Mitterrand University Hospital, 14 Rue Paul Gaffarel, BP 77908, 21000 Dijon, France; ludwig.aho@chu-dijon.fr

**Keywords:** hepatocellular carcinoma rupture, embolization, glue, cyanoacrylate, interventional radiology, bleeding control

## Abstract

The rupture of hepatocellular carcinoma (rHCC) is uncommon but causes sudden life-threatening bleeding. Selective transarterial embolization (STAE) is an effective bleeding-control option. The optimal embolic agent is unknown, and data on the use of cyanoacrylate glue are lacking. The objective of this study was to report the outcomes of cyanoacrylate-lipiodol embolization for rHCC. We retrospectively reviewed the files of the 16 patients (14 males; mean age, 72 years) who underwent emergency cyanoacrylate-lipiodol STAE at a single center in 2012–2023 for spontaneous rHCC. All 16 patients had subcapsular HCC and abundant hemoperitoneum. The technical success rate was 94% (15/16). Day 30 mortality was 19%; the three patients who died had severe hemodynamic instability at admission; one death was due to rebleeding. Liver enzyme levels returned to baseline by day 30. No rebleeding was recorded during the median follow-up of 454 days in the 10 patients who were alive with available data after day 30. Larger prospective studies with the collection of longer-term outcomes are needed to assess our results supporting the safety and effectiveness of cyanoacrylate-lipiodol STAE for rHCC. Randomized trials comparing this mixture to other embolic agents should be performed.

## 1. Introduction

Hepatocellular carcinoma (HCC) contributes to over 80% of primary liver cancers [1,2] and is the fourth source of cancer-related mortality worldwide, with 700,000 deaths each year [3,4]. HCC is usually, although not always, a complication of liver cirrhosis. The main risk factors are chronic hepatitis B or C, alcohol abuse, nonalcoholic fatty liver disease, and genetic susceptibility [3,5]. The spontaneous rupture of HCC (rHCC) is the third most common cause of HCC-related death after tumor progression and liver failure [6]. The diagnosis may be delayed in patients with previously undiagnosed chronic liver disease or HCC, particularly as the clinical manifestations are nonspecific, ranging from acute abdominal pain or distension to acute liver failure and hemorrhagic shock [7,8].

The Incidence of HCC varies across geographic regions [1,2], from less than 3% in occidental countries to up to 26% in Africa and Asia, where viral hepatitis is very common [8]. The mechanisms that cause rHCC are unclear but may include vascular injury related to the local aggressiveness of the lesion and venous congestion. Rupture is also more common when the tumor is located just under or near the capsule, with little or no surrounding liver parenchyma to provide a protective cushion (small room hypothesis) [8,9,10,11]. Other risk factors for rupture are large tumor size, tumor protrusion out of the liver contour, tumor location in the left lobe, liver cirrhosis, and underlying hypertension [8,9,10].

Multiphasic computed tomography (CT) is the best imaging modality for diagnosing rHCC [12]. Hemoperitoneum, subcapsular hematoma, or active bleeding is typically seen. Also highly suggestive is a peripheral mass exhibiting central necrosis and a protruding contour and producing a discontinuity on the liver surface (enucleation sign) [8,13].

The management of rHCC is aimed at ensuring patient survival by controlling the bleeding, with treatment of the tumor being provided only once the acute episode is controlled [14,15,16,17,18]. In the absence of surgical or radiological intervention, rHCC is often fatal due to bleeding and disease progression. Thus, in a 2022 study, the median survival in the 24 patients given only supportive treatment was 62 days [5]. Supportive treatment alone is associated with lower survival rates compared to surgical or radiological intervention. However, supportive treatment alone is reserved for patients deemed too ill to receive surgical or radiological intervention, for instance, due to severe liver failure or an advanced tumor stage [8,19,20]. When possible, either open surgery or transarterial embolization (TAE) is performed [21,22,23]. Emergency open surgery to resect the involved part of the liver has curative potential but is not feasible if the liver reserve is limited and/or marked liver cirrhosis is present [8,24]. TAE is less invasive and has been reported to stop the bleeding in 53% to 100% of patients [8,25,26]. A 2022 meta-analysis with 681 patients supported TAE followed by delayed hepatectomy, which was associated with significantly lower in-hospital mortality compared to emergency hepatectomy, although postoperative complication rates and both 1-year and 3-year survival rates were similar with the two strategies [21]. Another meta-analysis, with 974 patients, found significantly lower rates of both in-hospital mortality and complications after TAE or transarterial chemoembolization (TACE) compared to emergency hepatectomy, with similar 1-year survival rates [16].

The embolic agents available for TAE include mechanical devices such as plugs and coils, gelatin sponge particles [15], gel foam slurry mixed with iodized oil [25,27,28,29], microspheres [30], and polyvinyl alcohol particles [25,27]. Whether any of these agents is superior to the others for treating rHCC is unknown, and the choice often rests on the usual practice and preferences of each operator and institution [27]. Liquid glues such as cyanoacrylate, although widely used for various peripheral TAE procedures to treat tumors and other conditions, do not seem to have been evaluated in rHCC [31,32,33,34]. Cyanoacrylate is a radiolucent liquid that polymerizes upon contact with ion-rich fluids such as water, saline, and blood. Mixing cyanoacrylate with iodized oil (Lipiodol^®^ Ultra Fluid [LUF], Guerbet, Villepinte, France) provides opacification while also allowing modulation of the glue diffusion and polymerization kinetics.

The goal of this single-center retrospective observational study was to assess the outcomes of selective TAE (STAE) with cyanoacrylate–LUF mixtures in patients with rHCC.

## 2. Materials and Methods

### 2.1. Study Design and Patients

We retrospectively identified consecutive adults who underwent STAE with a cyanoacrylate-LUF mixture to treat rHCC at the Dijon-Bourgogne University Hospital (Dijon, France) between January 2012 and April 2023. The patients were identified by searching our imaging database using the indexing terms “hepatocellular carcinoma”, “HCC”, “rupture of hepatocellular carcinoma”, “hemoperitoneum”, “rupture”, “cyanoacrylate”, and “glue”. Patients in whom other embolic agents were used in addition to cyanoacrylate-LUF were eligible. Exclusion criteria were age younger than 18 years, post-traumatic rupture, bleeding from a tumor other than an HCC, uncertainty regarding the diagnosis of HCC, and STAE without cyanoacrylate-LUF.

### 2.2. Data Collection

For each patient, a single investigator (J.C.) used standardized forms to collect the following data from the medical files: age, sex, prior known health conditions including chronic liver disease and/or HCC, cardiovascular risk factors, antiplatelet and/or anticoagulant treatments, clinical presentation, CT findings (including the largest diameter and location of the bleeding HCC and the presence of other nonruptured HCC foci), vital status at last follow-up, and follow-up duration. The images acquired during the STAE procedure and subsequently were reviewed, as well as the corresponding reports, by the same experienced interventional radiologist (R.L.). The prothrombin ratio and INR at presentation were collected. Laboratory data were recorded at the time of STAE, then 48 h and 30 days later, including the hemoglobin level, platelet count, and levels of aspartate aminotransferase (AST) and alanine aminotransferase (ALT).

The results of the laboratory and imaging studies conducted during the subsequent follow-up were also recorded. After hospital discharge, the hepatologist and surgeon provided clinical follow-up and obtained laboratory tests as appropriate. The time to the first follow-up CT or magnetic resonance imaging (MRI) scan was usually three to six months and was at the discretion of the hepatologist. Subsequent imaging studies were performed as deemed appropriate by the multidisciplinary team in charge of each patient.

### 2.3. Diagnosis of Hepatocellular Carcinoma and of Rupture

All patients were admitted on an emergency basis with acute abdominal pain and/or evidence of internal bleeding. Emergent abdominal multiphasic CT was performed. The results were evaluated during an emergency multidisciplinary meeting involving the on-call interventional radiologist, emergency physician or hepatologist, anesthesiologist or intensivist, and hepatic surgeon. The images were assessed as detailed in the CT/MRI Liver Reporting and Data System (LI-RADS) v2018 [35]. In particular, the diagnosis of rupture relied on the following CT findings: peripheral or subcapsular tumor bulging out of the liver contour, discontinuity in the liver capsule, hemoperitoneum, subcapsular hematoma, parenchymal hematoma, active extravasation of contrast medium, and/or the enucleation sign [8,13,36]. The clinical features, notably the presence of cirrhosis, were also considered. Histological confirmation of HCC was obtained in some patients via examination of a percutaneous biopsy collected after bleeding control was achieved.

### 2.4. Selective Arterial Embolization Procedure

The decision to perform STAE was taken during the emergency multidisciplinary meeting. Hemodynamic stabilization was obtained before STAE. Fluid repletion, red-cell and platelet transfusions, and vasoactive amines were given as appropriate. The CT images were reviewed for details on liver arterial anatomy and identification of the tumor-feeding arteries, including any extra-hepatic feeders.

Local anesthesia was used except in intubated patients who received general anesthesia. The right common femoral artery was the most common approach. After arterial puncture under ultrasound guidance, a 5- or 6-Fr sheath was inserted. Angiograms of the celiac trunk and/or superior mesenteric artery and/or extrahepatic artery were obtained via a 5-Fr catheter. A microcatheter 2.0- to 2.7-Fr in caliber was then used for distal catheterization. Selective liver angiographies were performed to identify the tumor feeders and any sites of active bleeding. After selective microcatheterization of the target arteries, the microcatheter was flushed with 5% dextrose to remove ionic solutions that might cause early glue polymerization within the microcatheter lumen. When deemed appropriate, additional 5% dextrose was injected to fill the tumor bed, thereby enabling greater distality of embolization. The cyanoacrylate-LUF mixture was prepared shortly before the injection, using two 5-mL Luer-Lock syringes and a three-way stopcock. The main cyanoacrylates used were Glubran^®^2 (GEM, Viareggio, Italy) and MagicGlue^®^ (Balt, Montmorency, France). The cyanoacrylate-LUF ratio ranged from 1:2 to 1:6 according to microcatheter location and to whether the target vessel was proximal or distal. The mixture was injected under fluoroscopic guidance, using either free-flow or blocked-flow conditions (Figure 1). The effectiveness of the injection was assessed via fluoroscopy. The injection was stopped immediately when reflux into the microcatheter started to occur, and the microcatheter was then promptly removed (Figure 1).

A vascular closure device (AngioSeal^®^ or FemoSeal^®^, Terumo, Tokyo, Japan) was routinely placed at the femoral puncture site.

Each patient was monitored closely in the intensive care unit, hepato-gastroenterology department, or digestive surgery department. If rebleeding was suspected, multiphasic CT was to be performed routinely. If the results confirmed rebleeding, an emergency multidisciplinary meeting was to be held to assess treatment options, including repeat STAE.

### 2.5. Outcome Measures

The primary outcome measure was technical success, defined as total occlusion of the target vessels with absence of persistent bleeding on the angiography performed at the end of the STAE procedure.

The secondary outcome measures were rebleeding within 30 days after technical success as defined above, day 30 mortality, and day 30 rates of overall and major complications. Complications were classified according to Society of Interventional Radiology guidelines [37]: minor complications resolved without treatment but possibly required overnight hospitalization for observation, whereas major complications required specific care (admission and treatment or an unplanned increase in the level of care or prolongation of hospitalization) or resulted in permanent impairments or death.

### 2.6. Statistical Analysis

Quantitative variables were described as mean ± SD or median [interquartile range], and qualitative variables as numbers (%). The number of procedural failures was too small to allow a statistical analysis of risk factors for procedural failure.

## 3. Results

### 3.1. Patients and Tumors

Table 1 reports the main features of the 16 included patients. Nine highly experienced interventional radiologists performed the procedures. Only two patients had a known diagnosis of HCC at the time of the rupture. Table 2 reports the characteristics of the rupture. All 16 tumors were subcapsular. Contrast extravasation was visible in only half the patients.

### 3.2. SAE Characteristics

Table 3 describes the STAE procedures. Glubran^®^2 was used in all but one patient, who was treated with MagicGlue^®^, mixed with LUF. Nearly half the patients received an NBCA-LUF ratio of 1:5. For the first three procedures, the operators chose to add proximal gelatin sponge embolization via the lobar hepatic artery, although the cyanoacrylate-LUF embolization had stopped the bleeding. None of the 12 other patients (for those whose technical success was achieved) received embolic agents other than cyanoacrylate glue. A single liver segment was embolized in three-quarters of the patients (12/16 (75%)). Multisegment embolization was performed in four patients with large tumors invading several segments and supplied by multiple arteries.

### 3.3. Outcomes

STAE was technically successful in 15 (94%) patients. In the remaining patient, the bleeding was only partially controlled by STAE. Salvage proximal gelatin-sponge embolization via the common hepatic artery stopped the bleeding.

No major procedure-related complications occurred. No patient experienced acute liver failure. Transient post-embolization syndrome developed in seven (44%) patients.

Three patients died within two days after STAE. All three had been admitted with hypovolemic shock. One underwent technically successful STAE immediately after recovering from a cardiac arrest and died a few hours later. Another was the patient with technical STAE failure. The remaining patient experienced major, rapid clinical deterioration consistent with rebleeding but died before CT could be performed; this patient also had advanced lung malignancies. No other patients experienced manifestations consistent with rebleeding within 30 days of STAE. No other patients died within the first 30 days, and the day 30 mortality rate was 19%.

Three patients were lost to follow-up immediately after STAE. Median follow-up was 454 days (range: 1–1967 days). During follow-up in the ten survivors of day 30 with available data, none of the treated lesions showed abnormal neo-vascularization, denoting tumor progression.

Table 4 reports the laboratory data. Of the 16 patients, 8 required blood transfusions. The drop in the hemoglobin level from before baseline to 48 h after the procedure was negligible. The creatinine, alkaline phosphatase, and gamma-glutamyl transferase levels remained stable over time. Early increases in alanine and aspartate transferase levels were seen, but both enzymes returned to baseline values by day 30.

## 4. Discussion

STAE using cyanoacrylate glue was effective in stopping bleeding from rHCC in 15 of 16 patients included in this single-center retrospective study. Subsequently, a single patient experienced rebleeding. No major procedure-related complications were recorded. The day 30 mortality rate was 19%.

The prognosis of rHCC is bleak in the absence of surgical or radiological intervention, with nearly all patients dying within the first year, usually due to recurrent bleeding [15]. STAE or open surgery is associated with far lower rebleeding rates and better survival. However, we should bear in mind that supportive treatment alone is reserved for patients who have advanced HCC or other health conditions associated with a very short life expectancy.

STAE is a well-established treatment option for patients with rHCC and has superseded surgical hemostatic procedures such as hepatic artery ligation [17,18,25]. STAE is a minimally invasive method that does not require general anesthesia and eliminates the approach-related complications and infections that can occur after open surgery. A 2020 meta-analysis comparing selective TAE or TACE (n = 485) to emergency surgery (n = 489) demonstrated that embolization was associated with significantly fewer complications (odds ratio [OR], 0.36; 95% confidence interval [95%CI], 0.22–0.57; *p* < 0.0001) and in-hospital deaths (OR, 0.52; 95%CI, 0.29–0.94; *p* = 0.03). Bleeding control and 1-year survival were similar in the two groups [16]. We report only short-term survival (day 30) given our primary objective of assessing the effectiveness of STAE in achieving immediate hemostasis without rebleeding.

However, only one patient died during the follow-up (after the first 30 days). No patient received additional oncological treatment during the first 30 days of follow-up. Three patients underwent scheduled surgery (segmentectomy). Four patients underwent additional TACE, and one patient benefited from radioembolization.

Most of our patients had cirrhosis without clinically significant coagulopathy. The main cause of cirrhosis was alcohol abuse, compared to hepatitis B in populations studied in Asia [14]. The HCC was known before the rupture in only two of our 16 patients.

The very high technical success rate of 94% is in line with previous studies that used other embolic agents [14,15,38,39,40]. Rebleeding was rare, also in keeping with earlier data [16]. To our knowledge, this is the first study reporting the use of cyanoacrylate glue for emergency STAE in patients with rHCC. The best embolic in this setting remains unknown, and no randomized controlled trials have compared the available agents. Cyanoacrylate glue has been proven useful for treating nonruptured unresectable HCC [41,42]. The rapid polymerization of cyanoacrylate glue provides prompt hemostasis, a marked advantage in rHCC, given the risk of massive bleeding and hemodynamic instability [31,43,44]. Moreover, cyanoacrylate glue is effective in patients with coagulopathy, which is common in patients with HCC complicating cirrhosis [44]. Importantly, flow-directed microcatheterization can be used to ensure distal embolization even when the target arteries are difficult to catheterize due to tortuosity or slenderness. The addition of LUF slows the polymerization rate, and the cyanoacrylate–LUF ratio can be tailored to obtain the rate that is optimal for each specific situation [45,46]. The opacification provided by LUF enables monitoring of the flow of cyanoacrylate glue and the early detection of any nontarget embolization or reflux along the microcatheter. Contrary to microparticles and other liquid agents, cyanoacrylate glue adheres to the vessel wall, inducing substantial inflammation and remodeling that contribute to ensuring lasting lumen obliteration [46]. In a study of embolization to treat benign prostatic hypertrophy, cyanoacrylate glue was associated with shorter procedure and fluoroscopy times and with lower radiation doses compared to previous data obtained using microspheres [47]. Cyanoacrylate-LUF is safe, as shown by a 2021 meta-analysis including 574 patients managed for arterial gastrointestinal bleeding [48]. None of our patients experienced major complications. The absence of new or worsening liver failure can probably be ascribed to the selectivity of arterial embolization. Hepatic enzyme levels on day 30 were similar to those recorded at baseline. Finally, the day 30 mortality rate of 19% can be considered low, given the severity of rHCC. However, it should be noted that the optimal use of cyanoacrylate glue requires substantial experience.

The retrospective design and small sample size are the limitations of our study. Moreover, of the 16 patients, 3 were lost to follow-up immediately after STAE. However, rHCC is a rare event, and obtaining larger populations would require multicenter recruitment. Second, follow-up duration varied widely. The focus of our study was the effectiveness of cyanoacrylate-glue STAE in ensuring immediate hemostasis and good day 30 outcomes. Third, despite successful cyanoacrylate–glue embolization, a gelatin sponge was added for the first three patients as a precaution given the limited experience of the operators at the time. Fourth, the high technical success rate precluded an analysis of risk factors for technical failure, and we did not look for factors associated with day 30 survival.

## 5. Conclusions

STAE using cyanoacrylate glue was feasible, safe, and effective for the management of rHCC, according to our retrospective analysis of data from a small population. Larger studies with the prospective collection of longer-term data are needed, as well as randomized controlled trials comparing various embolic agents in rHCC.

## Figures and Tables

**Figure 1 jpm-13-01581-f001:**
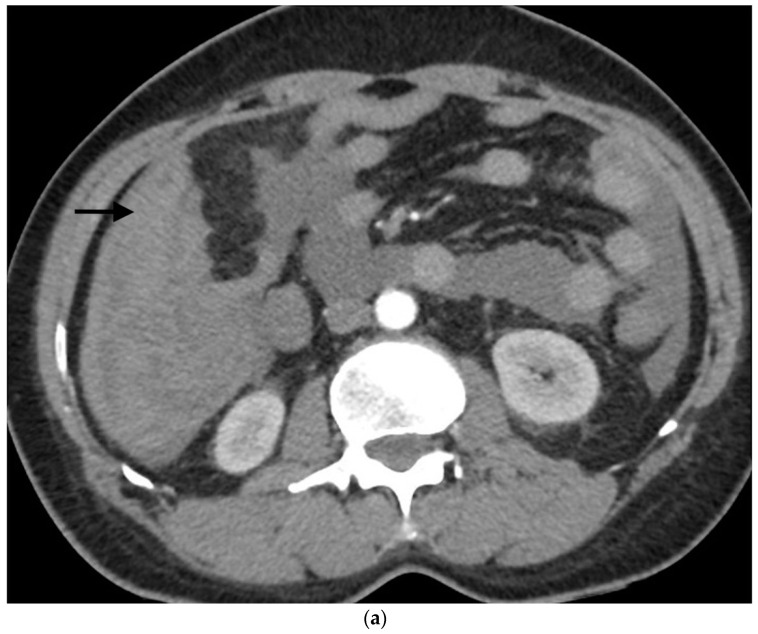
Example of ruptured hepatocellular carcinoma (rHCC) in a 52-year-old male who presented with signs of active intraperitoneal bleeding. (**a**) Unenhanced CT scan showing abundant perihepatic hemoperitoneum (arrow). (**b**) Enhanced CT images at the portal phase showing features highly suggestive of rHCC: large, heterogeneous, nodular hypervascular mass in a subcapsular location in segment VI (arrow), with intratumoral blush (arrow), on healthy liver. (**c**) Digital subtraction angiography via the superior mesenteric artery showing occlusion of the celiac trunk by median arcuate ligament syndrome, revascularization of the hepatic artery via the pancreaticoduodenal arcades, and hypervascular tumor (arrow). (**d**) DSA performed after selective microcatheterization of the main feeding arterial branch via the pancreaticoduodenal arcades, showing contrast-medium blush within the tumor (arrow). (**e**) Visualization of the cyanoacrylate/lipiodol mixture cast after selective injection in a 1:5 ratio in the main arterial branch (arrows). (**f**) Final angiographic control after glue embolization demonstrating cessation of active bleeding and complete devascularization of the tumor. (**g**) Enhanced MR imaging control at 2 months after embolization showing complete necrosis of the tumor. Complementary wedge hepatic resection was performed within 3 months. (**h**) MRI scan at 5 years after surgery showing hepatic surgical scar (arrow) with no tumoral recurrence.

**Table 1 jpm-13-01581-t001:** Main features of the 16 included patients.

Features	Data
Males/Females, n (%)	14 (88)/2 (12)
Age, years, mean ± SD	72.0 ± 9.3
Multifocal HCC, n (%)	7 (44)
Clinical presentation of rHCC, n (%)	
Acute abdominal pain	15 (94)
Symptoms of acute anemia	15 (94)
Hemodynamic shock	3 (19)
Chronic liver disease, n (%)	12 (75)
Liver cirrhosis	12 (75)
Child-Pugh A	4 (25)
Child-Pugh B	5 (38)
Child-Pugh C	3 (19)
NASH	1 (6)
Chronic alcohol abuse	8 (50)
HBV	1 (6)
HCV	0 (0)
Laboratory data at presentation	
Hemoglobin, g/dL, median [IQR]	9.6 [7.6–11.2]
Platelet count, G/L, median [IQR]	155 (127–172]
Platelets < 50 G/L, n (%)	0 (0)
Prothrombin ratio, median [IQR]	70 (59–82]
Prothrombin ratio < 50%, n (%)	1 (6.3)
Chronic antiplatelet/anticoagulant treatment, n (%)	
Anticoagulant therapy	2 (13) ^a^
Antiplatelet therapy	4 (25) ^b^

HCC, hepatocellular carcinoma; rHCC, ruptured HCC; NASH, nonalcoholic steatohepatitis; HBV, hepatitis B virus infection; HCV, hepatitis C virus infection. ^a^ Both patients were taking new oral anticoagulants (non-vitamin K antagonists). ^b^ All four patients were taking a single antiplatelet agent.

**Table 2 jpm-13-01581-t002:** Features of hepatocellular carcinoma ruptures (n = 16).

Features of the Tumors	Data
Location, n (%)	
Subcapsular	16 (100)
Side	
Left liver	6 (38)
Right liver	10 (62)
Largest diameter, cm	
Mean ± SD	5.2 ± 2.6
Median [IQR]	4.5 [3.6–10.5]
CT features	
Hemoperitoneum	14 (88)
Subcapsular hematoma	2 (13)
Parenchymal hematoma	2 (13)
Extravasation of contrast medium	8 (50)
Enucleation sign	0 (0)

n, number; SD, standard deviation; IQR, interquartile range; CT, computed tomography.

**Table 3 jpm-13-01581-t003:** Features of the selective transarterial embolization (STAE) procedures.

Features	Data
Number (%) of patients with	
1 embolized segment	12 (75)
2 embolized segments	2 (13)
3 embolized segments	1 (6)
Contrast medium extravasation during SAE, n (%) of patients	9 (56)
Number (%) of patients ^a^ with	
1 embolic agent injection site	7 (50)
2 embolic agent injection sites	4 (29)
3 embolic agent injection sites	3 (21)
Type of cyanoacrylate glue, n (%) of patients	
Glubran^®^2 ^b^	15 (94)
MagicGlue^® c^	1 (6)
NBCA/LUF ratio, n (%) of patients ^d^	
1:2	3 (20)
1:3	2 (13)
1:4	2 (13)
1:5	7 (47)
1:6	1 (7)
Technical success ^e^	15 (94)

NBCA, N-butyl cyanoacrylate; LUF: Lipiodol^®^ Ultra Fluid. ^a^ Data were missing for 2 patients. ^b^ NBCA mixed with a comonomer (GEM, Viareggio, Italy). ^c^ N-hexyl-cyanoacrylate (Balt, Montmorency, France). ^d^ Data were missing for 1 patient; the total exceeds 100% due to rounding. ^e^ defined as total occlusion of the target vessels with absence of persistent bleeding on the angiography performed at the end of the STAE procedure.

**Table 4 jpm-13-01581-t004:** Outcomes.

Outcomes	Data
Rebleeding within 30 days, n (%)	1 (6)
Died within 30 days, n (%)	3 (19)
Serum creatinine level, µmol/L, mean ± SD	
Just before STAE	125.0 ± 131.4
48 h after STAE	119.5 ± 99.9
Serum hemoglobin level, g/dL, mean ± SD	
Just before STAE	9.7 ± 2.1
48 h after STAE	9.3 ± 1.1
Serum AST level, IU/L, mean ± SD	
Just before STAE	89.8 ± 79.0
48 h after STAE	1260.7 ± 1389.4
30 days after STAE	118.1 ± 115.2
Serum ALT level, IU/L, mean ± SD	
Just before STAE	88.3 ± 78.0
48 h after STAE	760.8 ± 848.4
30 days after STAE	88.3 ± 63.3
Serum AP level, IU/L, mean ± SD	
Just before STAE	103.3 ± 41.7
48 h after STAE	124.8 ± 121.8
30 days after STAE	114.3 ± 36.1
Serum GGT level, IU/L, mean ± SD	
Just before STAE	248.1 ± 180.1
48 h after STAE	201.1 ± 163.2
30 days after STAE	208.3 ± 87.5

STAE, selective transarterial embolization; AST, aspartate aminotransferase; ALT, alanine aminotransferase; AP, alkaline phosphatase; GGT, gamma-glutamyl-transferase.

## Data Availability

All the study data are reported in this article.

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
