# Peer review of "Selective Arterial Embolization of Ruptured Hepatocellular Carcinoma with N-Butyl Cyanoacrylate and Lipiodol: Safety, Efficacy, and Short-Term Outcomes"

_jpm, 2023, doi:10.3390/jpm13111581_

Round 1
Reviewer 1 Report
Comments and Suggestions for Authors
The NBCA is currently the best emergency haemostatic method in ruptured hepatocellular carcinoma. 
It is particularly useful in rHCC when the catheter is far from the target artery and difficult to insert, as hemostasis can be achieved by injecting NBCA.
I have Two questions.
  1. Do you have any academic data showing that NBCA:LUF of 1:5 to 1:6 does not cause cerebral embolisation, pulmonary embolisation or other systemic circulatory concerns?
  2.I assume that the one-month post-CT was not performed, but in the three-month post-CT, were there any cases where the hemostatic NBCA-LPD injected into the ruptured area migrated and settled in the abdominal cavity, thoracic cavity or other areas?
Author Response
Response to Reviewer 1 comments
The NBCA is currently the best emergency haemostatic method in ruptured hepatocellular carcinoma. It is particularly useful in rHCC when the catheter is far from the target artery and difficult to insert, as hemostasis can be achieved by injecting NBCA.
Reply : Thank you very much for your comment.
I have Two questions.
- Do you have any academic data showing that NBCA:LUF of 1:5 to 1:6 does not cause cerebral embolisation, pulmonary embolisation or other systemic circulatory concerns?
Reply : Thank you very much for your comment. To our knowledge and after careful checking, we did not find any data in the literature regarding this purpose. From what we know, there is no reason to get pulmonary embolism using this dilution in the absence of big tumor shunts. Even in that case, glue is safer than particles because it is supposed not going through capillaries. Nothing has been added in the manuscript on that.

2.I assume that the one-month post-CT was not performed, but in the three-month post-CT, were there any cases where the hemostatic NBCA-LPD injected into the ruptured area migrated and settled in the abdominal cavity, thoracic cavity or other areas?
Reply : Thank you very much for your comment. No systematic 1-month CT scan was performed as mentioned. In patients with 3-month CT follow-up, we did not experience any specific out-tumor migration of the glue/lipiodol cast.
Reviewer 2 Report
Comments and Suggestions for Authors
The manuscript "Selective Arterial Embolization of Ruptured Hepatocellular Carcinoma with N-Butyl Cyanoacrylate and Lipiodol: Safety, Efficacy, an Short-term Outcomes" is a well-written retrospective case series. The background, methods, and results, along with the baseline patient data and outcomes, are complete and clear. The discussion appropriately covers the implications and decisions necessary to implement in practice.
The only significant improvement to the manuscript that I suggest is some editing of what might be a file-format error. In multiple locations a common appears to have been replaced with a ?. For example, see line 90 (Lipiodol?).
The authors could also consider choosing the single best case example rather than two. Twelve separate figure panels is overkill for a target audience that is well acquainted with bleeding on CT and DSA, and the post-embolization appearance of liver tumors/bleeding.
Author Response
Response to Reviewer 2 comments
The manuscript "Selective Arterial Embolization of Ruptured Hepatocellular Carcinoma with N-Butyl Cyanoacrylate and Lipiodol: Safety, Efficacy, an Short-term Outcomes" is a well-written retrospective case series. The background, methods, and results, along with the baseline patient data and outcomes, are complete and clear. The discussion appropriately covers the implications and decisions necessary to implement in practice.
Reply : Thank you very much for your comment.
The only significant improvement to the manuscript that I suggest is some editing of what might be a file-format error. In multiple locations a common appears to have been replaced with a ?. For example, see line 90 (Lipiodol?).
Reply : Thank you very much for your comment. It has been corrected through the entire manuscript, as suggested.
The authors could also consider choosing the single best case example rather than two. Twelve separate figure panels is overkill for a target audience that is well acquainted with bleeding on CT and DSA, and the post-embolization appearance of liver tumors/bleeding.
Reply : Thank you very much for your comment. Figure 1 has been removed, and Figure 2 has been kept as the best example and renumbered consequently as Figure 1, as suggested.
In addition, the references have been modified and renumbered as suggested by the Editor.